# OpenReview forum: "Unfiltered and Unseen: Universal Multimodal Jailbreak Attacks on Text-to-Image Model Defenses"
_ICLR.cc/2025/Conference — ICLR 2025 Conference Withdrawn Submission_

### Official Review · Reviewer_vLcA · 2024-10-30

**Soundness:** 3
**Presentation:** 3
**Contribution:** 2
**Rating:** 6
**Confidence:** 3

**Summary:**

This paper proposes U3-Attack, a universal multimodal jailbreak attack which aims to bypass the existing defense mechanisms placed on the generation of NSFW images via text-to-image models. The attack includes a paraphrase candidate set of circumvent text-based prompt filters and replaces sensitive words with context-free paraphrases to escape safety filters based on sensitive word detection. As or image defenses, U3-Attack uses adversarial patches which can bypass safety checkers with little dependence on model specifics, allowing for wide applicability to various T2I models and platforms. For experimental evaluation, the method was evaluated across models like Stable Diffusion, Leonardo.AI and Runway, with high success rates for the attack, exposing the vulnerabilities of current safeguards and highlighting the security gaps that need to be addressed.

**Strengths:**

1. The method is specified with detailed notation and provides logical reasoning of why each step was employed.

2. The paper conducted a comprehensive evaluation and experiments by comparing with existing methods of MMA-Diffusion and QF-Attack

**Weaknesses:**

Some early figures do not provide direct intuition of how the attack works. It provides some confusion and it makes the readers to carefully take a look at the figures.

**Questions:**

1. Could the paraphrase candidate set be adapted dynamically to adjust to evolving prompt filters?

2. How does the computational complexity vary for the image-modal attack with higher resolution models or increased patch size?

3. If I missed, does the attack affect model utility?

---

### Official Review · Reviewer_bvkc · 2024-10-30

**Soundness:** 3
**Presentation:** 3
**Contribution:** 2
**Rating:** 5
**Confidence:** 4

**Summary:**

This paper introduces a multimodal jailbreak attack targeting Text-to-Image (T2I) models, aiming to circumvent existing safeguards and generate NSFW content. The authors propose a universal attack that facilitates practical bypasses of prompt filters with minimal perturbations.

Additionally, the authors implement a two-stage adversarial patch generation strategy in the image modality. Then they show this approach is effective across both open-source models and popular online T2I platforms, including Stable Diffusion, Leonardo.Ai, and Runway.

**Strengths:**

Overall, this paper presents an attack that works well on both open-source models and popular online T2I platforms, thanks to a few clever techniques. From the results, it looks like their method not only outperforms two other baseline approaches but also does it faster.

The training pipeline is well-explained, and Figures 1, 2, and 3 are clear and easy to follow to understand the exact method.

**Weaknesses:**

1. I feel like the related work section is a bit lacking (even no related work section in the main paper). The discussion focuses mostly on MMA-Diffusion, but it could benefit from a broader comparison. Is MMA-Diffusion really the best benchmark to discuss here? From what I know, other works like Ring-A-Bell[1], JPA[2], and SneakyPrompt[3] have also tackled similar challenges with jailbreaking text-to-image models.

2. Without a more detailed comparison to related work, it’s hard to tell if the techniques in this paper are genuinely new or if they’re building on recent ideas. The specific challenges aren't very well explained either. For example, in the text-domain attack, it seems like a GCG-type approach should be effective—so what’s the exact challenge here? Additionally, there’s no ablation study for the techniques introduced in Section 2.2, which would help clarify their impact.

3. As for the results on online platforms, it might be more compelling to include more advanced APIs like DALL·E 2 and Midjourney in the evaluation.

[1] Ring-A-Bell! How Reliable are Concept Removal Methods for Diffusion Models?

[2] Jailbreaking Prompt Attack: A Controllable Adversarial Attack against Diffusion Models

[3] SneakyPrompt: Jailbreaking Text-to-image Generative Models

**Questions:**

When determining whether an image is malicious, are there any automated methods, or is it mainly done through manual observation? Like the results reported in Fig 9.

---

### Official Review · Reviewer_eGkq · 2024-11-02

**Soundness:** 3
**Presentation:** 3
**Contribution:** 2
**Rating:** 5
**Confidence:** 4

**Summary:**

In this paper, authors introduce U3- Attack, a novel multimodal jailbreak attack against T2I models that effectively circumvents existing safeguards to generate NSFW images. The U3-attack employs a sensitive word transform strategy to evade the prompt filter and utilizes a two-stage adversarial patch generation strategy to circumvent safety-checkers.

**Strengths:**

The U3-attack is universal. The jailbreak attack is suitable for many open-source models and online T2I models. The sensitive word paraphrasing method enables bypassing the prompt filters with minimal perturbation And the universal adversarial patch, obtained through a two-stage generation process, guarantees relevance to NSFW content and robustness against generation perturbations caused by content generation, even in non-edited regions.

**Weaknesses:**

The U3-attack strategy consists of two parts: text modality attack and image modality attack. The text modality attack is similar to the related work MMA; the improvement lies solely in replacing sensitive words with paraphrases based on similarity. The mechanism within the patch is not well explored.

**Questions:**

1、	Is the universal paraphrase set concrete and unchanged? Is it trained one-to-one based on sensitive words? Since the set is trained based on CLIP, is the text modality attack strategy robust against different encoders?
2、	Please explain why sensitive words are replaced with paraphrases, and whether the length of the paraphrase influences the performance of this module?
3、	The patch covers 7% of the input image. In the examples, it is set in the top-left corner of the image. I have concerns about whether the size and position of the patch influence performance.
4、	It is interesting that the patch experiences subtle degradation after passing through the T2I model, even when placed in the non-edited area of the image. It is said that this degradation results from the characteristics of lossy compression. Are there any experiments or something that can demonstrate this?

---

### Official Review · Reviewer_Gzbk · 2024-11-03

**Soundness:** 3
**Presentation:** 3
**Contribution:** 1
**Rating:** 3
**Confidence:** 4

**Summary:**

This paper introduces U3-Attack, a multimodal framework designed to circumvent safety guardrails T2I models, enabling the generation of NSFW content. This poses significant concerns regarding the potential misuse of these powerful generative models and highlights the need for more robust defense mechanisms.

**Strengths:**

- This paper addresses a critical safety issue: the misuse of T2Is through malicious or adversarial user inputs. As T2Is become more popular and widely deployed, this topic is increasingly important.

- The paper is well-structured and easy to follow.

**Weaknesses:**

**1. Lack of Originality and Technical Contributions.** The findings in this paper largely overlap with MMA-Diffusion [Yang et al., 2024]. For text modality, MMA-Diffusion uses the entire NSFW prompt as the optimization target, whereas U3-Attack targets specific NSFW words. In image modality attacks, MMA-Diffusion applies adversarial perturbations to input images to bypass safety checks, while U3-Attack employs adversarial patches. The modifications by U3-Attack seem more like adjustments at the setting level rather than methodological innovations.

**2. Insufficient Demonstration of Transferability and Generalizability to State-of-the-Art T2I Models.** The paper evaluates T2I models (SDv2.0, SDXL, SLD, Runway) in a black-box setting, all derived from the white-box SDv1.5. The authors should extend their attacks to other T2I models, such as DeepFloyd [a1] and Stable Diffusion 3 [a2], to demonstrate broader applicability and robustness.
[a1] https://github.com/deep-floyd/IF
[a2] https://huggingface.co/stabilityai/stable-diffusion-3-medium

**3. Insufficient Comparisons with Closely Related Work.** The QF-Attack is primarily designed to disable T2I functionalities rather than generate NSFW content. Therefore, Ring-A-Bell [a3] and P4D [a4] serve as more appropriate baselines for comparison.

[a3] https://arxiv.org/abs/2310.10012
[a4] https://arxiv.org/html/2309.06135v2

**4. Lack of Proper Citations.** Is the "Safety-Driven Loss Oriented Towards Safety Checker" mentioned in lines 191-218 being introduced for the first time in this paper?

**5. Effectiveness of U3-Attack Against Advanced Filters.** Can the proposed U3-Attack successfully bypass more sophisticated prompt filters, such as LlamaGuard [a5]?

[a5] LLamaGuard: https://arxiv.org/abs/2312.06674

**Questions:**

Please refer to the weakness section.

---

### Note · Authors · 2024-11-13

I have read and agree with the venue's withdrawal policy on behalf of myself and my co-authors.